# Perception of Bitter Taste through Time-Intensity Measurements as Influenced by Taste Modulation Compounds in Steviol Glycoside Sweetened Beverages

**Alex M. Pierce-Feldmeyer \*, Dave Josephson, Adrianne Johnson and Robert Wieland**

Mane Inc., 2501 Henkle Drive, Lebanon, OH 45036, USA

\* Correspondence: MINC.SensoryServices@mane.com; Tel.: +1-513-282-2653

**Abstract:** To limit sugar consumption and maintain sweetness levels in the diet, food and beverage developers often use high potency sweeteners (HPSs) as alternatives. Steviol glycosides are considered a consumer-friendly alternative but they are perceived to have a bitter taste accompanied by sweet and bitter lingering. Recently, taste modulators have been discovered that help to alleviate negative attributes like bitterness of HPSs. To show that taste modulation compounds (TMCs) decrease perceived bitterness associated with steviol glycosides, a trained descriptive panel ($n = 9$) performed a single-attribute time-intensity (TI) assessment over 2 min. Analysis of Variance (ANOVA) was used to analyze TI curves and curve parameters (AUC, Imax and Tmax). Principal components analysis (PCA) was also used to assess TI curves. Results showed that statistically significant results depended on the analysis method. Bitterness perception was shown to persist less over 2 min for steviol glycosides with TMCs when assessing raw scores and parameters. The same was not found using differences from control curves or weighted curves from PCA. These findings demonstrate that particular TMCs may subtly decrease perceived bitterness of steviol glycosides. However, business objectives of TMC use may dictate what kind of analysis method to use when analyzing perceived bitter perception of TMCs over time.

**Keywords:** taste modulation; sensory; bitterness; time-intensity; high potency sweeteners; steviol glycosides

---

## 1. Introduction

Product development in the food and beverage industry is currently focused on reducing added sugar which is highly associated with obesity risk and obesity-related illness, i.e., type II diabetes and cardiovascular diseases [1,2]. In an effort to lower the amount of added sugars in the diet, many companies use high potency sweeteners (HPSs), such as extracts of *Stevia rebaudiana* amongst others. The Food and Drug Administration (FDA) has approved six HPSs for use as food additives in the United States: Saccharin, aspartame, acesulfame potassium (Ace-K), sucralose, neotame and advantame [3]. Steviol glycosides received a generally regarded as safe (GRAS) status which does not require FDA approval as a food additive [4], and this status has driven consumer desire for steviol glycosides over many artificial sweeteners [5]. However, steviol glycosides are less acceptable than sucrose due to lost mouthfeel, bitterness, lingering aftertastes [6,7], astringency [6] and licorice or metallic off flavors [6,8,9]. Stevioside and Rebaudioside (Reb) A have been of most commercial interest, but other derivations of steviol glycosides have recently been promoted, including Reb B, D and M [10]. These preparations are claimed to have improved sensory characteristics such as "clean sweet" and milder bitter and licorice aftertastes [10–12]. These varieties have also been approved as GRAS

substances and have undergone in vitro metabolic studies, which concluded that the varieties were metabolized the same way as Reb A [10,13].

Recently, taste modulators were discovered as a tool to decrease negative attributes (bitterness, metallic, taste linger) of HPSs, meanwhile mimicking authentic sweet taste [8,14–20]. Taste modulator screening techniques have been developed to determine possible compounds that have positive effects on taste qualities, e.g., masking bitterness and enhancing sweetness. Thus far, studies have elucidated sweet taste modulators [17,18] and multi-attribute taste modulators [16]. Previous studies have found increased human perceptions of sweetness when positive allosteric modulators (PAMs) were added to sucralose [17] and sucrose [17,18]. A method to screen for taste modulators, comparative Taste Dilution Analysis (cTDA) enabled the discovery of alapyridaine, a multi-attribute taste modulator [16]. Alapyridaine was found to decrease the glucose threshold [21] as well as increase sweet, salty and umami intensity perceptions while having no effect on bitter or sour taste qualities [16]. Mechanisms of taste modulation are not entirely clear. Studies have suggested an interaction with domains of the sweet taste receptor heterodimer, specifically the transmembrane domain (TMD). This interaction has been suggested to increase activity and affinity to other domains like the so-called Venus flytrap domain (VFD) [17,18,20]. Furthermore, competitive inhibition has been shown to promote masking of bitter and astringent taste attributes of HPSs, reducing off-tastes inherent to HPSs (e.g., steviol glycosides). The hydrophobic regions of the receptor membranes are blocked with hydrophobic taste modulating compounds (TMCs), that may bind with greater affinity than the HPS, therefore reducing off-taste and lingering of bitter and astringent taste attributes, while not affecting sweet enhancement [22]. Thus, competitive inhibition at the receptor site and/or sequestering of hydrophobic bitter molecules are yet another mechanism to reduce the off-taste of multi-attribute HPS sweeteners [22,23].

The process of proving that taste modulation is meaningful to consumers has been difficult. Sensory studies thus far have been correlative to non-human analytical findings. Studies have used detection thresholds [14,19,24], comparative Taste Dilution Analysis (cTDA) [16], sensory-directed fractionation [25], perceived sweetness intensity [17], paired comparisons [18], equivalence studies [17] and consensus profiling with difference testing [26]. However, time varying changes of bitter taste were not fully considered as an indication of sweet taste modulation. By reducing a solution's perceived bitterness, sweetness perception increases [27]. Further, time-intensity profiles of modulated beverages are critical for thorough HPS sweetener evaluation [8]. It should be noted that tastes are multimodal and integrated. In response to modulating one or more taste actives, a range of consequential taste-modifying effects ensues, influencing the integrated perceptions of all five tastes and off-perceptions.

In order to begin showing meaningful changes in HPSs using taste modulators, a single attribute Time-Intensity (TI) method was used to evaluate bitterness intensity of steviol glycosides with and without TMCs. For this study, TMCs are defined as single molecules that do not exhibit any taste on their own, nor do they impart any aroma at low levels. The purpose of this study was to develop a sensitive method and analysis for the screening of potential bitter reducing TMCs for diet beverages using steviol glycosides. This is considered a first step in building a sensory platform for measuring multimodal and integrated taste effects of TMCs. This approach is designed to lead to new taste modulating insights, which will help to increase the understanding of how to improve steviol glycoside sweetened foods and beverages using TMCs.

## 2. Materials and Methods

### 2.1. Participants

All procedures in this study were approved by an Institutional Review Board (IRB) (Protocol #: Pro00028683; Advarra, Columbia, MD, USA). Informed written consent was acquired from all participants. A descriptive panel (*n* = 9) with over 400 h of training and experience was used for data collection. Panel members were invited to be on the panel after undergoing extensive taste and aroma sensitivity testing.

### 2.2. Materials

*Stevia rebaudiana* (Rebaudioside A 97%, Nascent, New York, NY, USA) was dosed into filtered water at 275 ppm to mimic 5% (*w/v*) sucrose sweet perception [28]. This control solution was then used to dose TMCs A, B and C. Three TMCs were included in this study: TMC A, a lactam [molecular weight (mw) < 100 daltons], TMC B, a lactam (mw < 120 daltons) and TMC C, a lactone (mw < 220 daltons) (Mane, Inc., Lebanon, OH, USA). By this methodology, experiments used the precise control solution with and without a single TMC (Table 1). Filtered water was used to dissolve TMCs prior to dosing which were added to the control solution at 10 ppm. All samples were mixed on a stir plate (Corning Stirring Hot Plate, VWR, Radnor, PA, USA) at medium speed (500 digital read out) for 4 min. Samples were prepared 24 hours before the test and were stored in amber glass bottles (Piramal Glass—USA, Inc., Dayton, NJ, USA) at refrigeration temperature (4–6 °C) prior to portioning. It should be noted that sucrose was not compared directly to these samples to avoid a context effect [29].

**Table 1.** Samples assessed for bitterness time intensity over 2 min.

| Samples | Usage Level (%) [1] |
|---|---|
| Rebaudioside A 97% (Control) | 0.0275 |
| Rebaudioside A 97% + TMC A [2] | 0.0275 + 0.001 |
| Rebaudioside A 97% + TMC B [3] | 0.0275 + 0.001 |
| Rebaudioside A 97% + TMC C [4] | 0.0275 + 0.001 |

[1] Usage level reflects that of the sweetener and taste modulation compound (TMC) in water. [2] A lactam with a molecular weight <100 daltons. [3] A lactam with a molecular weight <120 daltons. [4] A lactone with a molecular weight <220 daltons.

### 2.3. Temporal Evaluation

Approximately 3 fluid ounces (oz.) of sample was portioned into black 4 oz. cups (Gordon Food Service, Grand Rapids, MI, USA). Cups were labeled with random three-digit codes for each product and capped with a lid. Samples were stored at 4 °C and were allowed to equilibrate to room temperature (21 °C) for 2 hours prior to testing. Descriptive panelists (*n* = 9) calibrated with bitter references (quinine at 5, 10, 15, 20 and 25 ppm) that ranged from low to high on a 10-point intensity scale, which was used for the evaluations. Quinine was used as the bitter reference due to steviol glycosides activating the same bitter taste receptor, hTAS2R14 [30,31], and due to the panel agreeing that it aligned well with their bitter perceptions when tasting steviol glycosides.

Prior to sample testing, training consisted of aligning and training the panelists based on their perceived bitterness intensity for steviol glycosides (control). The bitterness reference scale consisted of quinine levels representing 1.5, 3, 5, 7 and 8.5 on the bitter intensity scale that was used and referenced during evaluations. These references were also used to calibrate prior to each test session. Additionally, descriptive panelists practiced Time-Intensity (TI) measurements over 2 min on a computer interface using Fizz Network Software Acquisitions Biosystèmes 2.51A (Dijon, France) to aid in familiarity and comfort with data collection prior to testing.

Descriptive panelists evaluated the control and samples with TMCs in fully enclosed booths under white light. Panelists were instructed to simultaneously taste the sample and swirl three times,

while using the computer mouse to click on the line scale their perceived intensity for bitterness (0 = none, 10 = high). After swirling the sample in their mouth, panelists expectorated the sample and continued rating. Expectoration was the preferred method for the panelists to prevent the potential for a satiety effect. The intensity scoring was controlled by the panelist using the computer mouse; moving the mouse to the right increased the perceived bitterness score and moving the mouse to the left decreased the perceived bitterness score. Panelists were prompted to continuously rate until the end of 2 min. The panel was not informed that the sensation would end; panelists were allowed to end the 2 min with a palpable bitterness rating due to the persistent linger of steviol glycoside sweeteners.

Preliminary testing helped to identify that unflavored, carbonated water, oyster crackers and milk chocolate were the most effective palate cleanser to help with bitter taste and linger associated with steviol glycosides. A minimum of 3 min was used to cleanse the palate between samples. This procedure occurred until all samples were assessed. Samples were randomized using a Latin Square design.

*2.4. Statistical Analysis*

Traditional TI curve parameters, including area under the curve (AUC), intensity maximum (Imax) and time to maximum intensity (Tmax) were calculated for control and variant samples. One-way Analysis of Variance (ANOVA) was used to compare curve parameters. For the raw bitterness intensity curves, a mixed model repeated measures ANOVA was performed with panelist as a random effect; sample, time and sample*time interaction as fixed effects; and with bitterness scores as the dependent variable. Additionally, the same analysis method was used on the difference from control bitterness scores as the dependent variable. Lastly, principal component analysis (PCA) was used to generate weighted average curves [32,33]. These curves were statistically compared using a mixed model repeated measures ANOVA with sample, time and sample*time interaction as fixed effects with factor scores as the dependent variable. Pairwise comparisons of the raw average bitterness scores, the difference from control bitterness scores and the weighted average bitterness scores derived from PCA were tested using Fisher's Least Significant Difference (LSD) method. An alpha level of 0.05 was considered significant and an alpha level of 0.10 was considered marginally significant. All statistical analyses were performed using the Sensory package in XLSTAT (Version 2018.6, Addinsoft, Inc., Long Island City, NY, USA).

## 3. Results

Panelists ($n$ = 9) rated bitterness intensity of steviol glycoside samples with and without potential TMCs over 2 min. After reviewing the data, one panelist was deemed to be an outlier due to opposing interpretations of sample intensities compared to the rest of the panel. Therefore, this panelist was removed from the data set.

*3.1. Comparison of Raw Average TI Curves among Control and Variant Samples with TMCs*

Average bitterness curves of steviol glycosides with TMCs were lower over time than steviol glycosides alone ($p < 0.001$). However, effect sizes were small when compared to the control for TMC A ($p$ = 0.126), TMC B ($p$ = 0.115) and TMC C ($p$ = 0.104) (Figure 1).

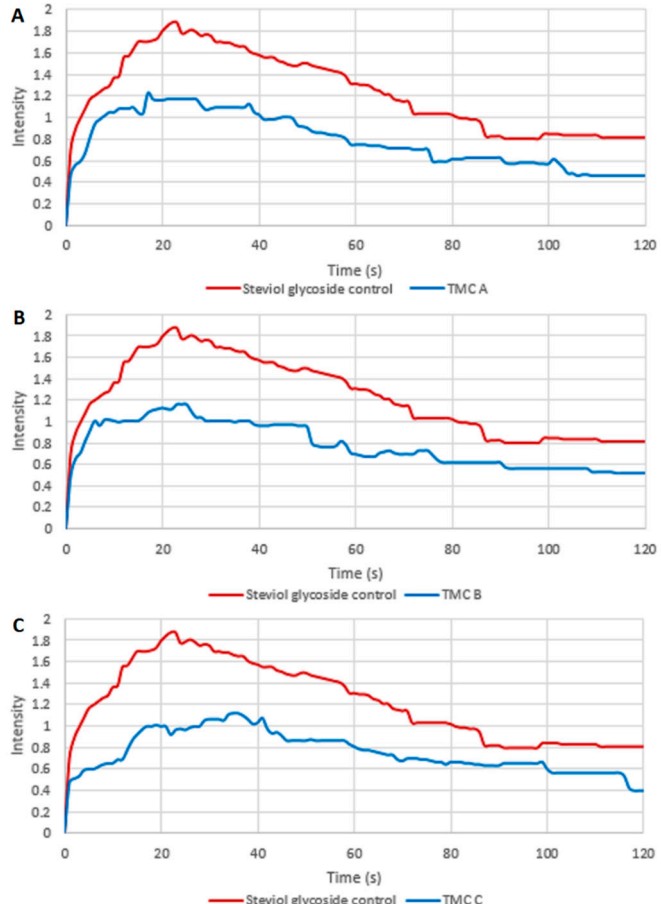

**Figure 1.** Bitterness intensity average scores over 2 min for the steviol glycoside control and variants (**A**) TMC A, (**B**) TMC B and (**C**) TMC C. Sample was a significant fixed effect in a mixed model ANOVA ($p < 0.001$). Minimal effect sizes were determined with Fisher's Least Significant Difference (LSD) test for TMC A ($p = 0.126$), TMC B ($p = 0.115$) and TMC C ($p = 0.104$) compared to the control.

### 3.2. Comparison of Raw Average TI Curve Parameters among Control and Variant Samples with TMCs

The TI parameters derived from the average curves were the Imax, Tmax and AUC (Table 2). There were marginally significant effects of AUC ($p = 0.066$) between the variants and the control. After making pairwise comparisons, TMC A ($p = 0.033$), TMC B ($p = 0.028$) and TMC C ($p = 0.024$) had significantly lower AUCs compared to the control. Significant effects of Imax were also found ($p = 0.049$) between TMC A ($p = 0.029$), TMC B ($p = 0.042$) and TMC C ($p = 0.011$) when compared to the control. Tmax did not significantly differ between control and variants with TMC A, B or C ($p > 0.05$).

**Table 2.** Statistical significance of mean ± standard error (SE) of time intensity parameters for the steviol glycoside control and variants.

| Sample | Imax [1] | Tmax [2] | AUC [3] |
|---|---|---|---|
| Rebaudioside A 97% (Control) | 2.03 [a] ± 0.41 | 19.89 [a] ± 4.97 | 147.52 [a] ± 40.56 |
| Rebaudioside A 97% + TMC A | 1.43 [b] ± 0.34 | 21.89 [a] ± 9.00 | 94.23 [b] ± 26.43 |
| Rebaudioside A 97% + TMC B | 1.48 [b] ± 0.30 | 22.44 [a] ± 9.73 | 92.52 [b] ± 25.00 |
| Rebaudioside A 97% + TMC C | 1.32 [b] ± 0.30 | 30.00 [a] ± 9.24 | 90.97 [b] ± 26.73 |
| *p*-value | $p = 0.049$ | $p = 0.534$ | $p = 0.066$ |

[1] Maximum intensity score over the 2 min time duration of the TI analysis. [2] Time (seconds) of maximum intensity score over the 2 min time duration of the TI analysis. [3] Area under the time intensity curve. In each column, values with the same letter did not differ statistically; values with different letters differed significantly ($p < 0.05$).

### 3.3. Comparison of Average Bitter Difference TI Curves among Control and Variant Samples with TMCs

The perceived bitterness scores of the control were subtracted from the perceived bitterness scores of the variant at each time point in order to generate difference curves. Bitter difference scores were significantly different ($p > 0.001$) (Figure 2). The bitter difference scores (variant–control) were marginally lower for TMC A ($p = 0.063$) and TMC B ($p = 0.055$), whereas bitter difference scores were significantly lower for TMC C ($p = 0.049$) compared to control.

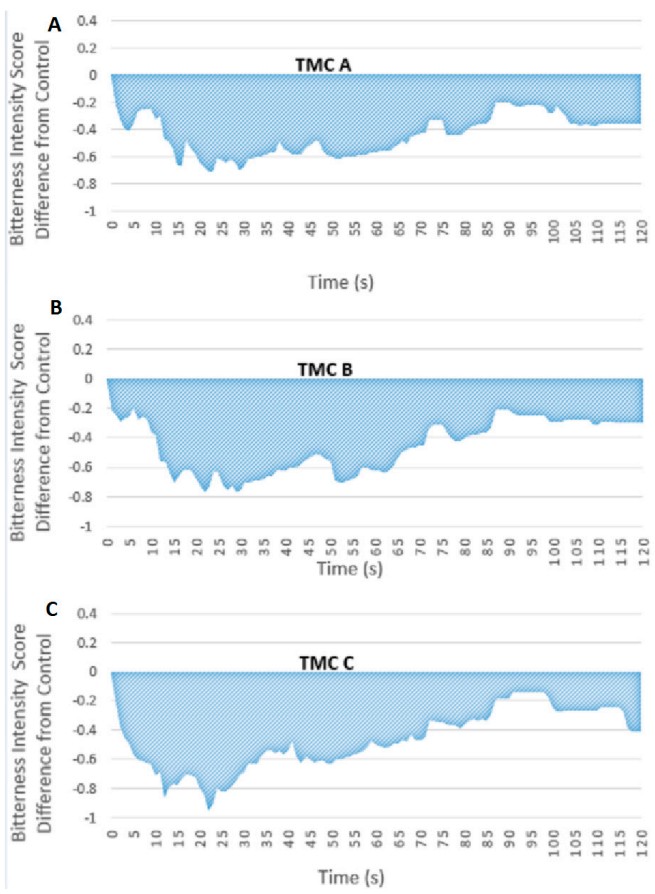

**Figure 2.** Bitterness intensity difference from the control for (**A**) TMC A, (**B**) TMC B and (**C**) TMC C. A negative change indicates that average bitterness intensity of the samples with TMCs was less than the average bitterness intensity of the control (steviol glycosides). Pairwise comparisons indicated a marginally significant difference in perceived bitter difference from the control for TMC A ($p = 0.063$) and TMC B ($p = 0.055$). TMC C ($p = 0.049$) significantly increased the perceived difference in bitterness from the control.

### 3.4. Comparison of PCA-Generated Average TI Curves among Control and Variant Samples with TMCs

A PCA was performed to generate non-centered average curves as described by Dijksterhuis et al. [33]. No sample effect was observed ($p = 1.00$) (Figure 3).

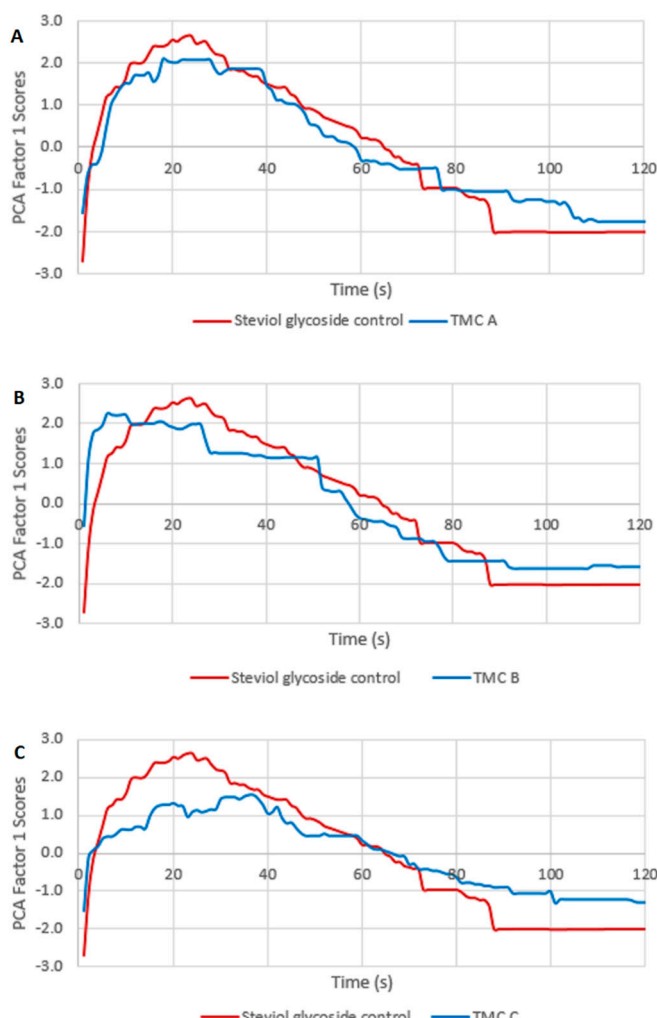

**Figure 3.** PCA-derived average curves for the steviol glycoside control and steviol glycoside variants with (**A**) TMC A, (**B**) TMC B and (**C**) TMC C. There was no significant difference between the control and variant samples ($p > 0.05$).

## 4. Discussion

Descriptive panelists rated bitter intensity of steviol glycosides with and without TMCs over 2 min. This method was used to test TMCs for their potential to reduce bitterness associated with steviol glycosides. Multiple TI statistical techniques to assess TMC efficacy were compared.

### 4.1. TI Parameters Show Significant Differences between the Steviol Glycoside Control and Steviol Glycosides with TMCs

A significantly lower intensity maximum suggests that peak bitterness is less for steviol glycoside samples with TMCs. A marginally lower AUC suggests bitterness over the entire 2 min is less than the control. There were no significant differences in Tmax, suggesting bitterness does not peak at different times than the control. However, all samples with TMCs had a longer time until maximum bitterness was reached (Table 2). This demonstrates that the time course of bitter intensity does not differ significantly with the presence of TMCs, but that the overall bitter intensity of samples with TMCs was significantly lower than the control. Previous studies have not collected TI bitterness curves for steviol glycosides with TMCs, but Hillmann et al. [25] has shown sweet enhancement from 5-acetoxymethyl-2-furaldehyde in a sucrose solution with TI curves. These results suggest an associated binding mechanism of the TMC which may directly compete with binding of the bitter-tasting actives.

It is necessary to note that the peak bitterness intensity (Imax) recorded throughout the study averaged towards the beginning of the 10-point scale, at approximately 2 which equates to a low intensity. This is likely due to the 97% purity level of Rebaudioside A tested in this study. A higher purity Rebaudiosde A results in less bitter-tasting steviol glycoside isomers within the extract. This steviol glycoside was of 97% purity, reducing the intensity of bitter off-tastes, and thus lessening peak bitter perceptions (perceived to be better tasting).

Derived TI parameters have primarily been used to assess differences among TI curves [34–41]. TI parameters, however, reduce TI curves to several indices, possibly losing information that differentiates samples. Additional statistical methods were tested to determine whether sample differences were clearer when using each entire curve.

*4.2. Raw Average Bitterness Scores Were Reduced with Added TMCs*

Average bitterness curves of steviol glycosides with TMCs were lower over time than steviol glycosides alone ($p < 0.001$). However, effect sizes were small when compared to the control for TMC A ($p = 0.126$), TMC B ($p = 0.115$) and TMC C ($p = 0.104$) (Figure 1). We hypothesized that reduced bitterness would be notable when TMCs were added to steviol glycosides in water. However, bitterness is not the only off-note attributed to steviol glycosides and stronger differences may have been seen if other attributes were measured. Steviol glycosides, specifically Rebaudioside A, has been known to elicit an astringent, metallic and licorice off-flavor that is undesirable [8]. These bitterness measurements cannot account for other off-notes or tastes that could affect the overall perception of steviol glycosides and thus further differentiate samples with TMCs. Other studies have shown decreased bitterness [16], as well as enhanced sweetness [16–18,25], suggesting stronger differences from TMCs may have been found if multiple taste or off-taste effects were measured over time.

*4.3. Bitter Difference Scores Shows Minimal Effects of TMCs*

Bitter difference scores were calculated to normalize signatures associated with panelist TI curves. All TMCs showed, on average, a bitterness reduction, but differences were not significant (Figure 2). Small differences may be due to a lack of TMC efficacy on steviol glycoside bitterness, however it is possible that bitter differences were more difficult to differentiate in the context of other distracting HPS attributes: sweetness, sweet linger, astringency, metallic and licorice off notes [8]. Additionally, small differences may be because sweet beverage solutions are not inherently bitter and thus may be harder to differentiate when they are perceived at relatively low intensities. TMCs are also of interest because of their ability to affect perceptions subtly, ideally not changing the flavor profile associated with a beverage. Small changes may not suggest significant differences, but statistically insignificant effects may still be meaningful to consumers that consume beverages with HPSs regularly. This will be important for specific business propositions. The context of the test will need to be incorporated when interpreting the analyses.

*4.4. PCA-Generated Average Curves Show Minimal Differences among Samples*

PCA was also used to generate average curves for these samples as it has been considered the "best summary of the data" [42]. This is an alternative way to average TI curves [32] in order to account for more variance in raw data and lessen individual differences [33]. Differences among the PCA-generated curves were not significant, suggesting that the samples did not differ significantly on perceived bitterness over time (Figure 3). This study demonstrates differing sensory results when applying alternative sensory analyses (TI curves parameters, entire TI curves, changes from baseline and PCA-generated curves) suggesting the need for additional efforts to reliably and consistently measure the taste modifying effects perceived through TMCs.

Previous TI analyses using PCA found small visual differences for chewing gum [42], lager beers [32] and scotch malt whiskey [43]. Other studies have used PCA to further interpret TI parameters [31,44,45]. It is clear that PCA is beneficial when desiring a deeper understanding of

TI parameter relationships and for averaging curves that are less influenced by panelist signatures. However, sensitivity for detection of subtly different samples may be harder to determine using this method due to its averaging technique.

The averaging technique may benefit TI curves generated from untrained panelists. However, by normalizing the data, we may also lose important information from raw data. Descriptive panelists were all tested for bitter blindness, but some panelists are super-tasters, which could explain curve distortions [46] and test dynamics we would expect to see in the commercial environment. These genetic differences may actually be helpful when assessing subdued differences in bitterness since high sensitivity is needed for differentiation. It might be more important to use the raw data for interpretation of trained panel data, because those with higher bitter sensitivity would be more functional. The best type of analysis to use may depend on the type of panelists (trained or untrained, panel size, etc.) performing TI measurements. Statistical analysis may also depend on whether the testing is for commercialization. The business context and objective may dictate how critical the reduction of either Type I or Type II error will be.

*4.5. Multimodal Taste Modulation*

There are a multitude of attributes associated with the distaste of HPSs like steviol glycosides, including bitter, metallic, licorice off-tastes, differences in the temporal nature of the sweet system (delayed onset and linger) and differences in mouthfeel [47]. The development of novel sensory methodologies will need to consider the multidimensional nature of taste. Humans incorporate bitter, metallic and sweet sensations (among others) as an integrated taste perception, not as separate perceptions of tastes or off-notes. This integration allows humans to taste components simultaneously, which combine to create an overall perception. When consuming foods or beverages, the overall perception is compared to a "standard" of expectation for this food or beverage. In this case, HPSs are used to replace sucrose, making sucrose the standard to which HPSs are compared. By focusing on reduction of bitterness with added TMCs, it is perceivable that studies may be able to capture shifts in taste profiles that appear more sugar-like in sweetness onset and aftertaste (less bitter). Though these measures lack explicit indications of improved quality and enhancement of sweet perception, by using an important correlate to sugar-like quality (bitter reduction), we can begin to better understand if and how TMCs may enhance the sugar-like quality of steviol glycosides. Future studies will focus on methods that incorporate the integration of all relevant sweetener attributes that affect sugar-like perceptions.

*4.6. Limitations*

This investigation's preliminary studies indicated that no differences were perceived when the panelists were evaluating sweetness over time. Other research studies have shown increased sweet perception in the presence of molecules which modulate taste, but many of these molecules also may add taste or aroma which are known as Flavorings with Modifying Properties (FMPs) [16–18,25,48,49]. The molecules tested here are without excess taste or aroma attributes and are therefore not considered FMPs. This distinction may indicate different mechanisms, suggesting the importance of incorporating multiple sensory methods to form a more complete understanding of sweetener changes. Recently, more methods have been developed that address multiple attributes over time, including Temporal Check-All-That-Apply (TCATA) and Temporal Dominance of Sensation (TDS) that could be useful in determining additional attributes that may be affected by TMCs. Future studies will aim to collect more TI data on multiple attributes affected by TMCs.

These results are only conclusive for TMCs affecting model systems of steviol glycosides in a water solution and not a full beverage formula with proper additives. Future studies will aim to collect more data to support TMC effects over time, as well as effects notable in more representative matrices found in the market, like diet carbonated beverages. Additionally, these studies will also aim to use hedonic scales to determine whether subtle changes in bitterness are meaningful to diet

beverage consumers. This will allow for broader interpretations of sample differences in the context of consumers. It is critical to understand whether these differences are meaningful to consumers and to understand the magnitude of difference when considering a business proposition context and consumer-relevant risk [50].

## 5. Conclusions

Our findings show that TI bitterness ratings may serve as a valuable screening tool for potential TMCs. It is important to consider the utility of such a method in a business context when choosing the statistical analysis. Both the panel size and training, as well as the business proposition will dictate the most appropriate analytical method to use. Future studies will include TMC assessment in diet beverage matrices in which diet beverage consumers will be utilized in order to evaluate multiple attributes and beverage acceptability. These studies will enable a better understanding of TMC effects.

**Author Contributions:** A.M.P.-F. participated in the study conceptualization, design, coordination, administration, statistical analysis and writing of the manuscript; D.J. participated in the study conceptualization, design, investigation, provided the resources and reviewed the manuscript for submission; A.J. approved the study for funding, participated in the study conceptualization, investigation, and reviewed the manuscript for submission; and R.W. approved the study for funding, participated in the study conceptualization, supervised the study, and reviewed the manuscript for submission.

**Funding:** This research was funded internally by Mane, Inc.

**Conflicts of Interest:** The authors of the study declare a conflict of interest. This study was funded privately by Mane, Inc. Mane Inc. employs the authors of the study. This played no role in the design of the study; in the collection, analyses, or interpretation of data; in the writing of the manuscript, or in the decision to publish the results.

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
