# Peer review of "Perception of Bitter Taste through Time-Intensity Measurements as Influenced by Taste Modulation Compounds in Steviol Glycoside Sweetened Beverages"

_beverages, doi:10.3390/beverages5030052_

Round 1

Reviewer 1 Report

Manuscript is well written and presented

The authors address need revision. What city and country is Mane Inc?

In both figures 1 and 2 the control Rebaudioside A 97% was not included. This should be included.

Line 329: Remove 'can' after 'ratings' and replace with 'may'. The limitations of the study are many

The 47 references cited has not been written as per guide. Authors should consult guide and re-write accordingly 

Author Response

Manuscript is well written and presented. The authors thank you for your comment.

The authors address need revision. What city and country is Mane Inc? The authors thank you for your comment. The address has been added and only the correspondence email remains.

In both figures 1 and 2 the control Rebaudioside A 97% was not included. This should be included. The authors thank you for your comment. In Figure 1, the authors believe the control is included and is referenced in the legends as “Steviol glycoside control.” For Figure 2, the curves are “difference from control” and therefore the control is embedded in this derivation of the data. The plots indicate the bitter score change from control for each variant’s scores.

Line 329: Remove 'can' after 'ratings' and replace with 'may'. The limitations of the study are many The authors thank you for your comment and ‘can’ has been replaced with ‘may.’

The 47 references cited has not been written as per guide. Authors should consult guide and re-write accordingly. The authors thank you for your comment. The guide has been referenced, changes have been made and references have been reformatted to fit the journal style. Thank you.

Reviewer 2 Report

Dear Authors,

The manuscript "Perception of Bitter Taste Through Time-Intensity Measurements as Influenced by Taste Modulation Compounds in Steviol Glycoside Sweetened Beverages" is an interesting work that reported the steviol glycosides consumer acceptance. It is a good paper, well written, easy to read. Background is good and provides sufficient coverage. A clear, objective and interesting discussion about the results, highlighting the relevance and the importance of this study was presented. In general, the description of the methods was made with many details. A proper discussion of the results obtained was carried out. The given approach considerably enriches the knowledge about the characterization of steviol glycoside.

However, please respond to the following, minor comments:

Tables and Figures were presented with adequate quality, but not in accordance with the recommendations of the publishing house. Please read the instructions for authors again.

Line 41 & 73: Abbreviations used in the work should be explained at their first use also in the content of the work  (‘HPSs’, ‘TMCs’)

Line 92: There is: ‘sucrose sweet perception ([24] Espinoza et al. 2014).’ There should be: ‘sucrose sweet perception [24].’

Line 113: Please remove ‘Meyerhof et al. 2009;’

Line 196: Add Surname before [29]

Line 214: same remark, [21]

Author Response

Dear Authors,

The manuscript "Perception of Bitter Taste Through Time-Intensity Measurements as Influenced by Taste Modulation Compounds in Steviol Glycoside Sweetened Beverages" is an interesting work that reported the steviol glycosides consumer acceptance. It is a good paper, well written, easy to read. Background is good and provides sufficient coverage. A clear, objective and interesting discussion about the results, highlighting the relevance and the importance of this study was presented. In general, the description of the methods was made with many details. A proper discussion of the results obtained was carried out. The given approach considerably enriches the knowledge about the characterization of steviol glycoside.

The authors thank you for your review

However, please respond to the following, minor comments:

Tables and Figures were presented with adequate quality, but not in accordance with the recommendations of the publishing house. Please read the instructions for authors again. The authors thank you for your comment. The Figure descriptions were placed underneath each figure. Any other changes that needed to be made were unknown to the authors.

Line 41 & 73: Abbreviations used in the work should be explained at their first use also in the content of the work  (‘HPSs’, ‘TMCs’) The authors thank you for your comment. Changes have been made in the main text for high potency sweeteners and taste modulation compounds. Thank you.

Line 92: There is: ‘sucrose sweet perception ([24] Espinoza et al. 2014).’ There should be: ‘sucrose sweet perception [24].’ The authors thank you for your comment. Proper citation changes were made.

Line 113: Please remove ‘Meyerhof et al. 2009;’ The authors thank you for your comment. ‘Meyerhof et al. 2009’ has been removed.

Line 196: Add Surname before [29] The authors thank you for your comment. The surname et al. has been added.

Line 214: same remark, [21] The authors thank you for your comment. The surname et al. has been added.

Thank you very much for your comments.

Reviewer 3 Report

Author affiliation - since everyone come from the same company, perhaps consider collating it to one? Corresponding emails are usually added by MDPI directly when the paper is accepted.

Introduction

First paragraph is hanging and isn't so relevant to the paper topic, consider deleting it.

Second paragraph, consider expanding latest steviol glycosides development (e.g. Reb M).

Third paragraph, more information is needed here to add value to the reader. Perhaps it is worth adding the mechanism on how these modulator works.

Section 2.3, Line 110 - how is the panel exactly temporally calibrated here? More details needed

Section 3, Line 155 - what crossover effect? Elaborate.

Figure 1 caption. How exactly the Fisher LSD used here? Is it per time second period comparison or average or AUC? Elaborate.

Figure 3. How did exactly the stats test is carried out here? It's rather confusing on how this has been carried out for the NPTIC curves.

Other parameters in TI curve remained unexplored here, how about the rate of increase/decrease (SIMInc/SIMDec)? I think it would be interesting to investigate this considering the temporality aspect of TMC.

Minor comments:

Line 92 & 113, etc., referencing is incorrect. Please refer to author's guideline.

Line 102, Table caption needs to be re-done, a Table is supposed to make sense even in isolation.

Author Response

Author affiliation - since everyone come from the same company, perhaps consider collating it to one? Corresponding emails are usually added by MDPI directly when the paper is accepted. The authors thank the reviewer for this comment and have reduced the reference to Mane, Inc. to “1,” for each author.

Introduction

First paragraph is hanging and isn't so relevant to the paper topic, consider deleting it.

The authors thank the reviewer for this comment and have deleted the first paragraph.

Second paragraph, consider expanding latest steviol glycosides development (e.g. Reb M). The authors thank the reviewer for this comment. More information has been added regarding recent developments, sensory and metabolic effects, and regulatory news regarding new preparations of steviol glycosides. Thank you.

Third paragraph, more information is needed here to add value to the reader. Perhaps it is worth adding the mechanism on how these modulator works. The authors thank the reviewer for this comment. Clear mechanisms have not been defined, especially because each taste modulator may act differently. However, more background was added from work by Servant et al. (2010, 2011) discussing possible mechanisms of some known modulating systems which affect the affinity and activity of domains on sweet taste receptors.

Section 2.3, Line 110 - how is the panel exactly temporally calibrated here? More details needed. The authors thank the reviewer for this comment. Temporal calibration of bitter references was elaborated describing the references and the scores they are associated with. This scale was used prior to testing and was available to reference during testing.

Section 3, Line 155 - what crossover effect? Elaborate. The authors thank the reviewer for this comment and have changed the wording for this effect. It was meant that the panelist disagreed with the other panelists in the interpretation of sample intensity difference.

Figure 1 caption. How exactly the Fisher LSD used here? Is it per time second period comparison or average or AUC? Elaborate. The authors thank the reviewer for this comment. The description of the particular scores being compared was added. Average bitterness scores were compared in order to incorporate the most information from each curve.Pairwise comparisons of the raw average bitterness scores, difference from control bitterness scores and the weighted average bitterness scores derived from PCA were tested using Fisher’s Least Significant Difference (LSD) method.”

Figure 3. How did exactly the stats test is carried out here? It's rather confusing on how this has been carried out for the NPTIC curves. The authors thank the reviewer for this comment. These statistics were the same after a principle component analysis which generated factor scores that became ‘weighted average scores’ of the TI profiles. This method was adopted from van Buuren (1992) and Dijksterhuis et al. (1994). These references are made in the document.

Other parameters in TI curve remained unexplored here, how about the rate of increase/decrease (SIMInc/SIMDec)? I think it would be interesting to investigate this considering the temporality aspect of TMC. The authors thank the reviewer for this comment. The purpose of these statistical analyses was to go beyond TI parameters. The authors noticed that often in the literature only TI parameters are used instead of ‘whole curve’ analyses, potentially losing important information from the sensory testing session and panelist interpretation. The major TI curve parameters such as Imax, Tmax, and AUC were incorporated for comparison purposes to ‘whole curve’ statistical methods. The analyses for this method were performed as a way to better utilize the information from the curve and reduce the use of extraneous TI parameters. The following paragraph was included in the discussion section.

"Derived TI parameters have primarily been used to assess differences among TI curves ([30-37]). TI parameters, however, reduce TI curves to several indices, possibly losing information that differentiates samples. Additional statistical methods were tested to determine whether sample differences were clearer when using each entire curve."

Round 2

Reviewer 3 Report

The authors had addressed all the comments accordingly.

Author Response

Attached are the most up to date versions, in response to the Academic Editor's minor revisions. 
